# Association between health indicators of maternal adversity and the rate of infant entry to local authority care in England: a longitudinal ecological study

Rachel Jane Pearson [ORCID],[1] Matthew Alexander Jay [ORCID],[1] Linda Petronella Martina Maria Wijlaars [ORCID],[1] Bianca De Stavola,[1] Shabeer Syed,[1] Stuart John Bedston,[2] Ruth Gilbert[1]

[1]Population, Policy and Practice Research and Teaching Department, UCL Great Ormond Street Institute of Child Health, London, UK
[2]Centre for Child and Family Justice Research, Department of Sociology, Lancaster University, Lancaster, UK

**Correspondence to**
Rachel Jane Pearson;
rachel.pearson@ucl.ac.uk

## ABSTRACT

**Objective** Infants enter care at varying rates across local authorities (LAs) in England, but evidence is lacking on what is driving these differences. With this ecological study, we aimed to explore the extent to which adversity indicated within women's hospitalisation histories, predelivery, explained the rate of infant entry into care.

**Methods** We used two longitudinal person-level data sets on hospitalisations and entries to care to create annual measures for 131 English LAs, between 2006/2007 and 2013/2014 (April–March). We combined these measures by LA and financial year, along with other publicly available data on LA characteristics. We used linear mixed-effects models to analyse the relationship between the outcome—LA-specific rate of infant entry into care (per 10 000 infants in the LA population) — and LA-specific percentage of live births with maternal history of adversity-related hospital admissions (ie, substance misuse, mental health problems or violence-related admissions in the 3 years before delivery), adjusted for other predictors of entry into care.

**Results** Rate of infant entry into care (mean: 85.16 per 10 000, SD: 41.07) and percentage of live births with maternal history of adversity-related hospital admissions (4.62%, 2.44%) varied greatly by LA. The prevalence of maternal adversity accounted for 24% of the variation in rate of entry (95% CI 14% to 35%). After adjustment, a percentage point increase in prevalence of maternal adversity—both within and between LAs—was associated with an estimated 2.56 (per 10 000) more infants entering care (1.31–3.82).

**Conclusions** The prevalence of maternal adversity before birth helped to explain the variation in LA rates of infant entry into care. Preventive interventions are needed to improve maternal well-being before and during pregnancy, and potentially reduce risk of child maltreatment and therefore entries to care. Evidence on who to target and data to evaluate change require linkage between parent–child healthcare data and administrative data from children's social care.

## Strengths and limitations of this study

► This is the first study to examine the relationship between local authority variation in rate of entry into care in England and parental health indicators that are associated with diminished parenting capacity.

► In our analysis, we used a breadth of information on mothers and children derived from longitudinal, person-level Hospital Episode Statistics records, and our outcome was derived from the Children Looked After return database containing all entries into care in England.

► Our estimates of local authority prevalence of live births with history of maternal adversity-related hospital admission (ie, substance misuse, mental health or violence-related admissions up to 3 years before delivery) are likely to be underestimated due to poor recording, reporting or recognition of these problems.

► Due to the lack of national data linkages between mother-to-baby linked healthcare data and children's social care outcomes in England, we could not examine whether children born to women with a history of maternal adversity-related hospital admission were more likely to be placed into care during infancy.

## INTRODUCTION

England has experienced unsustainable increases in the number of children in public care in recent years, leaving its children's social care sector overburdened.[1 2] One in five children entering public care are infants (aged under 1 year), and the rate of infant entry into care has increased by almost 20% since 2010/2011.[1] Similar patterns and increases in infant entry to care are taking place in other countries, including Scotland, the USA, New Zealand and parts of Australia. There is also marked regional variation in rates of entry across each of these settings,[3–7] with limited evidence on what drives differences.[8 9] The most frequently documented entry reason among infants entering care in England is abuse and neglect (ie, child

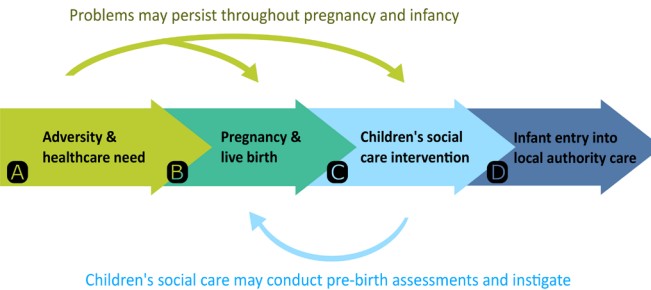

Problems may persist throughout pregnancy and infancy

Adversity & healthcare need **A**

Pregnancy & live birth **B**

Children's social care intervention **C**

Infant entry into local authority care **D**

Children's social care may conduct pre-birth assessments and instigate early intervention during pregnancy

**Figure 1** A hypothetical (simplified) pathway from parental adversity to infant entry into care. (A) Substance misuse, domestic violence and abuse, and mental health problems not only affect day-to-day functioning but can also lead to serious and complex healthcare needs. (B) These forms of adversity can also affect the capacity to parent and may result in harm to the unborn child or infant. Also, pregnancy, birth and caring for an infant place additional stress on parents, which can exacerbate experiences of adversity. (C and D) Where children are at a significant risk of harm, children's social care services have the power to apply for a court order to receive the child into care or may otherwise receive a child into care where to do so is in the child's best interests and the parents do not object.

maltreatment).[1] Maltreatment in infancy is linked to poorer physical, intellectual and behavioural development in childhood, and to substance misuse, risky sexual behaviour, domestic violence and abuse, self-harm, and poorer health outcomes later in life.[10 11] Developing our understanding of the drivers of geographical variation in the prevalence of maltreatment and demand for child protection intervention in infancy is, therefore, a key public health issue.

The responsibility for placing infants into care in England rests with the 152 local authorities (LAs) (ie, local governmental bodies with responsibilities of providing community services, such as schools, housing and social care) who have a statutory duty to safeguard children.[12] Past studies have shown that entry into care is driven by a complex ecological framework of risk factors relating to children and their families, the local community, service thresholds and capacity, and wider society.[13–15] These studies also highlight strong associations between parental adversities, such as mental health problems or deprivation, and child maltreatment or entry into care. Experiences of substance misuse, domestic violence and mental health problems are common among parents whose children are subject to child protection in England.[16] Parents who experience these forms of adversity are also more likely to have serious and complex healthcare needs,[17] as well as being more likely to struggle to meet their child's needs and to maltreat their child (figure 1).[13 14] In Western Australia and Manitoba, Canada, analyses of linked record-level administrative health and social care data showed that maternal mental health, substance-related or assault-related hospital admissions were associated with a higher relative risk of child entry into care.[18] There is also evidence that children whose parents experience

these forms of adversity are more likely to be subject to repeated child protection intervention.[19 20] Therefore, in this study, we examined the extent to which LA-specific prevalence of health indicators for maternal adversity prior to birth explained variation in rate of infant entry into care among LAs in England. With this work, we aim to inform national and local policy strategies to mitigate long-lasting harm to children arising from serious parental adversity and diminished capacity to parent.

As English children's social care data are not yet linked at the child level to longitudinal healthcare data, we conducted an ecological analysis. We have explored the relationship between LA-specific rates of maternal hospitalisation before birth due to substance misuse, domestic violence and abuse, and mental health problems and LA variation in the rate of infant entry into care. We also accounted for a range of other LA-level risk factors for entry into care, including LA-specific prevalence of maternal deprivation, births to teenage mothers and community violence.

## METHODS

### Study design

This longitudinal, ecological analysis used yearly aggregate measures between the 2006/2007 and 2013/2014 financial years (April–March) for 131 out of a possible 151 (as of April 2006) English LAs, derived from several data sources. We excluded 20 LAs from our analysis for having too few live births or poor data quality in at least one measure in one or more years (table 1).

### Outcome

Our outcome was the annual incidence rate of first entry into care during infancy (ie, under 1 year old) per 10 000 infant residents in the LA. The longitudinal Children Looked After (CLA) data set, provided by the Department for Education, was used to determine the numerator for each LA, with the Office for National Statistics population estimates being used to define the denominator.[21] CLA is a statutory data collection collated by the Department for Education and contains record-level information on all children in care in England, based on annual submissions from each LA. CLA includes information on child demographics (eg, age, sex, ethnicity and so on) and episodes in care (eg, episode dates, legal status, placement type and so on).[22] We used a longitudinal extract of CLA, which contained episode-in-care-level information for all children who first entered care during infancy between 1 April 2005 and 31 March 2014, to derive our outcome. Infants who entered care for any reason were included, with the exception of infants who first entered care under respite arrangements (ie, an agreed series of short-term breaks, typically provided for children with complex healthcare needs). For further information on this extract, see online supplementary appendix page 3.

### Explanatory measures

We derived several other explanatory measures, selected to capture further 'demand-side' risk factors (ie, local

**Table 1** Measures used in this study

| Measure type | Measure | Temporal coverage | Description | Data source(s) | Limitations |
|---|---|---|---|---|---|
| Outcome | Rate of infant entry to care. | 2006/2007 to 2013/2014 | The number of children who first enter care during infancy, per 10 000 infants in the LA population, by financial year of first entry. | CLA return (linked by LA to Office for National Statistics midyear infant population estimates). | If a child in care is transferred to the care of another LA, or is adopted but later returns to care, they will receive a new identification number. This could lead to double counting; however, LA transfers and adoption breakdowns are uncommon. |
| Descriptive (ie, not used in modelling) | Number of singleton live births recorded in HES APC. | 2006/2007 to 2013/2014 | Number of singleton live births recorded in HES APC where maternal age is non-missing and there is at least one English LSOA recorded in maternal HES APC record in the look-back period. | HES APC. | We only had access to data where date of birth was non-missing; therefore, births where maternal age was missing are not captured in this analysis. Two LAs were excluded for having fewer than 100 singleton live births in at least one financial year between 2006/2007 and 2013/2014. |
| Explanatory | LA population size. | 2006/2007 to 2013/2014 | Number of individuals living in the LA. | Office for National Statistics midyear population estimates. | The Office for National Statistics only provide information on the accuracy of estimates from 2013 onwards. |
| Explanatory | % of live births with maternal history of ARA. | 2006/2007 to 2013/2014 | % of singleton live births recorded in hospital where the mother had at least one ARA* in the 3 years prior to delivery. | HES APC. | Up to 20 ICD-10 codes are available per episode of inpatient care in HES APC (up to 14 in 2006/2007); however, the number of codes recorded likely differs among hospitals. This may result in underestimation of this measure in some LAs. |
| Explanatory | % of live births where mother <20 years old. | 2006/2007 to 2013/2014 | % of singleton live births recorded in hospital where the mother was less than 20 years old at delivery. | HES APC. | There were very few quality issues with birth dates in the HES APC extract (eg, <10 or >50 years old at delivery). |
| Explanatory | % of live births where maternal LSOA history within the 10% most deprived LSOAs in England. | 2006/2007 to 2013/2014 | % of singleton live births recorded in hospital where the mother lived in an LSOA that was one of the 10% most deprived LSOAs in England (according to the 2010 IMD) within the 3 years prior to delivery. | HES APC (linked by LSOA to 2010 IMD measures). | The LSOA used to derive maternal deprivation could be up to 3 years out of date at time of delivery. In addition, where women with multiple LSOAs recorded in the look-back period, each LSOA was linked to the 2010 IMD deciles and the minimum decile of deprivation (ie, most deprived) from all LSOAs recorded was selected. |

Continued

**Table 1** Continued

| Measure type | Measure | Temporal coverage | Description | Data source(s) | Limitations |
|---|---|---|---|---|---|
| Explanatory | % of live births where child has a complex chronic condition. | 2006/2007 to 2013/2014 | % of singleton live births with mother–baby linkage where the child had a congenital anomaly—identified where a congenital anomaly-related ICD-10 code† was recorded in the child's HES APC record within the first 2 years of life or recorded on a death certificate before the age of 5 years old (to capture children whose congenital anomaly diagnosis was not captured at birth or who were diagnosed later in life). | HES APC. | Information on children with congenital anomalies was only available for births with mother–baby record linkage. Therefore, this measure was calculated using only singleton live births with linkage available. A further nine LAs were excluded as they were missing mother–baby record linkage for more than 35% of singleton live births in at least one financial year between 2006/2007 and 2013/2014. |
| Explanatory | % of live births with low birth weight. | 2010/2011 | % of singleton live births where child had a low birth weight—identified where recorded birth weight <2500 g or a low birth weight-related ICD-10 code (P05.0, P07.0 or P07.1) was recorded in child's HES APC record within 7 days of delivery. | HES APC. | There is considerable variation in quality of birthweight recording by hospitals. Where birth weight was missing in the delivery record but mother–baby linkage was available, we looked for recorded birth weight in the child's birth record and for ICD-10 codes related to low birth weight. The quality of birthweight recording also varied from year to year and therefore we decided to use data only from the 2010/2011 year (the midpoint of our study period). A further nine LAs were excluded as they were missing a recorded birth weight in the maternal or child (where available) HES APC record at birth for more than 35% of singleton live births between April 2010 and March 2011. |
| Explanatory | % of dependent child households with lone parent. | 2011 | % of households with dependent children (ie, children aged 0–15 years old), where there is a single parent. | Census 2011 (Table LC1109EW). | |

Continued

**Table 1** Continued

| Measure type | Measure | Temporal coverage | Description | Data source(s) | Limitations |
|---|---|---|---|---|---|
| Explanatory | Rate of violent crime (per 100 LA residents). | 2010/2011 | The number of violence against the person offences, based on police-recorded crime data, per 100 people residents in the LA. | Public Health England Fingertips (Indicator 11202). | This does not capture violent crimes not reported to, or recorded by, the police. In addition, rate of violent crime in city centres with few residents (such as the City of London) may be inflated as there will be large numbers of people commuting into these areas who are not counted in the population denominator. |

*We defined history of ARA as any episode of admitted patient care related to substance misuse, mental health problems (including self-harm) or exposure to violence in the look-back period, determined by several non-mutually exclusive lists of ICD-10 codes.[27–30]
†Diagnoses of congenital anomalies were identified using a subset of Feudtner *et al*'s[31] ICD-10 code list (ie, all Q codes).
ARA, adversity-related hospital admission; CLA, Children Looked After; HES APC, Hospital Episode Statistics Admitted Patient Care; ICD-10, International Statistical Classification of Diseases and Related Health Problems 10th Revision; IMD, Index of Multiple Deprivation; LA, local authority; LSOA, lower-layer super output area.

service need) for child maltreatment and entry into care, alongside our exposure of interest (table 1). These measures were chosen based on evidence,[13–15] data availability and quality, and interpretability and were used to account for potential confounders in our statistical analyses.

### Longitudinal patient-level data on hospital admissions

We used the Hospital Episode Statistics Admitted Patient Care (HES APC) database to derive our exposure of interest and other maternal and child characteristics near to birth (table 1). HES APC captures all National Health Service-funded hospital admissions in England and covers 97% of all births and 98%–99% of all hospital admissions.[23] HES APC consists of records by episode of inpatient care, each with a pseudonymised patient identifier attached, allowing researchers to longitudinally link inpatient episodes over a patient's life course. Each inpatient episode record captures up to 20 (14 before April 2007) patient diagnoses using the International Statistical Classification of Diseases and Related Health Problems 10th Revision (ICD-10).[24]

We created an extract that included all singleton live births recorded in HES APC between 1 April 2006 and 31 March 2014. We derived a look-back period for each birth, which included all recorded maternal inpatient episodes within the 3 years prior to delivery to maximise identification of risk factors prior to birth. We excluded births where all recorded maternal areas of residence over the look-back period—identified via the lower-layer super output area (LSOA) code—were non-English or missing, as we required an English LSOA to derive maternal deprivation status (via the 2010 English Index of Multiple Deprivation).[25] Using Harron *et al*'s[26] longitudinal linkage of HES APC for mothers and babies in England we also

had access to the child's HES APC records for 96% of births in the extract.

Our exposure of interest—the proportion of live births with maternal history of adversity-related hospital admission (ARA)—was defined as any hospital admission related to substance misuse, exposure to violence or mental health problems during the look-back period using mutually non-exclusive ICD-10 code lists (for ARA ICD-10 codes, see online supplementary appendix page 6).[27–30] We also derived four further explanatory measures: (1) the proportion of live births where maternal age was under 20 years old; (2) the proportion of live births where maternal LSOA history was within the 10% most deprived LSOAs in England; (3) the proportion of live births with low birth weight; and (4) the proportion of live births where the child was diagnosed with a congenital anomaly in early childhood (table 1).[31] For further information on this extract, see online supplementary appendix pages 4–5.

### Publicly available data

We obtained all other yearly LA figures for risk factors from publicly available data. We used data from Census 2011 to derive the percentage of dependent child households with a lone parent (ie, single-parent households) and used LA-specific rates of violent crime published by Public Health England as a proxy measure for prevalence of LA violence.[32 33] We also included LA population size in our set of explanatory measures, from the Office for National Statistics midyear population estimates,[21] to account for differences between LAs with larger and smaller resident populations.

### Longitudinal modelling

We analysed the relationship over time between the LA-specific, yearly rate of infant entry into care (which

was approximately normally distributed) and LA-specific, yearly percentage of singleton live births with a maternal history of ARA, using linear mixed-effect models with restricted maximum likelihood estimation (for histograms of the outcome by financial year, see online supplementary appendix page 7). LA-specific percentage of live births with maternal history of ARA was modelled as a time-varying covariate between 2006/2007 and 2013/2014 to allow us to examine whether the association under study changed over time. However, the estimated model coefficient for this time-varying covariate could be interpreted as either (1) the effect of a unit increase in this covariate within the same LA or (2) the effect of a unit increase in this covariate across different LAs. Therefore, we sought to disaggregate these two effects by replacing the original LA-specific variable with two variables: (1) the mean percentage of live births with maternal history of ARA over the study period for each LA (ie, the between-LA effect), and (2) the difference, within LAs, between the original LA-specific yearly values and the LA-specific mean (ie, the within-LA effect). The coefficient for the first variable captures the between-LA effect and the coefficient for the second the within-LA effect.[34] We used Wald $\chi^2$ tests to test the null hypothesis that these two effects were equal. All other LA-level risk factors for entry into care were included in models as non-time-varying variables using data from 2010/2011 only (ie, midpoint of study period) as we observed minor variations and this improved model parsimony.

We fitted five models: (1) a ('null') model with only financial year as the explanatory variable; (2) a model with financial year and both LA-specific mean and mean-centred maternal ARA prevalence among live births; (3) a model which included all explanatory measures (as listed in table 1), including mean and mean-centred maternal ARA prevalence; (4) a model which included all explanatory measures and the original maternal ARA prevalence; and (5) model 4 with an interaction between financial year and maternal ARA prevalence. Models 1–4 included random intercepts for LA and random slopes for financial year. Model 5 included only random intercepts for LA as more complex random-effects structures did not converge.

The assumption of normality for the level 1 residuals of each model was checked using quantile-quantile plots and histograms, and we inspected fixed-effect parameter estimates and SEs for inflated values that would be symptomatic of multicollinearity. We used the Akaike information criteria to assess relative goodness of fit and all five models were checked for implausible predicted values. We also performed model-based parametric bootstrap (with 10 000 simulations) to estimate the proportion of variation in the outcome explained by the whole model (ie, by both fixed effects and random effects) and by only the fixed effects using formulas for a conditional and marginal pseudo-$R^2$ value, respectively.[35–37] We reported the median marginal and conditional pseudo-$R^2$ values from the bootstrapped samples, along with 95% CI (using the percentile method). All data management and analyses were carried out using R V.3.5.1.

### Patient and public involvement
No patients or members of the public were involved in the development or analyses of the current study. However, this study was carried out to inform a wider piece of work: linkage of administrative family court and healthcare data to better understand the healthcare need and health service use of women involved in care proceedings in England, particularly those with a mental health illness. For this larger project, we attended two regular group meetings of mental health service users to discuss the project and the data linkage between administrative family court data and mental health service records in England. We also held a focus group with women who have been involved in care proceedings to discuss the study in more detail and gain feedback on the project objectives. These public involvement sessions have helped us to strengthen our research plan, improving the relevance of future findings to the population under study.

## RESULTS
Table 2 summarises the LA characteristics for the study cohort by financial year. The median number of residents per LA increased over time (56 455 residents in 2006/2007 vs 60 426 in 2013/2014), while the median number of live births (3288 in 2006/2007 vs 3415 in 2013/2014) and the median percentage of live births where maternal LSOA history was within the 10% most deprived English LSOAs (14.46% in 2006/2007 vs 14.72% in 2013/2014) remained stable. Across all 131 LAs, the median rate of infant entry into care (72.76 per 10 000 in 2006/2007 vs 90.14 in 2013/2014) and the median percentage of live births with maternal history of ARA (2.73% in 2006/2007 vs 7.01 in 2013/2014) increased over time. LA-specific rates for both these measures varied substantially each year between LAs. The median percentage of live births where the child had a congenital anomaly also increased over time (1.64% in 2006/2007 vs 1.93% in 2013/2014), although LA variation decreased over time (min–max: 0.60%–3.34% in 2006/2007 vs 1.03%–3.22% in 2013/2014). Both the median percentage of live births to mothers under 20 years old (7.01% in 2006/2007 vs 4.33% in 2013/2014) and the LA variation in this measure (min–max: 1.14%–14.50% in 2006/2007 vs 0.87%–8.49% in 2013/2014) decreased over time. There was variation between LAs in the proportion of live births with low birth weight (min–max: 4.22%–9.94%), the rate of violent crime (0.52–3.17 per 100 residents) and the proportion of dependent child households with a lone parent (9.78%–30.94%).

### Modelling the association between rate of infant entry into care and prevalence of maternal ARA before birth
Figure 2A displays the point estimates and 95% CIs from models 1–5 for the coefficients of the time-varying

**Table 2** Local authority characteristics

| LA characteristics, median (min–max) | 2006/2007 (n=131) | 2007/2008 (n=131) | 2008/2009 (n=131) | 2009/2010 (n=131) | 2010/2011 (n=131) | 2011/2012 (n=131) | 2012/2013 (n=131) | 2013/2014 (n=131) |
|---|---|---|---|---|---|---|---|---|
| Number of singleton live births recorded in HES APC | 3288 (323–16 076) | 3416 (299–16 687) | 3454 (295–16 847) | 3516 (309–16 722) | 3552 (264–16 734) | 3550 (298–17 132) | 3440 (333–16 932) | 3415 (303–16 438) |
| Rate of infant entry to care (per 10 000) | 72.76 (0.00–240.00) | 66.25 (8.83–184.70) | 72.99 (4.62–197.18) | 79.19 (0.00–280.26) | 81.89 (4.51–195.74) | 90.16 (10.26–253.56) | 93.05 (0.00–318.51) | 90.14 (13.94–269.50) |
| LA population size | 56 455 (8000–313 364) | 56 783 (8140–315 580) | 57 453 (8114–317 556) | 58 339 (7995–318 654) | 58 570 (8079–320 618) | 59 039 (8031–322 743) | 60 307 (7809–324 341) | 60 426 (7795–326 092) |
| % of live births with maternal history of ARA | 2.73 (0.52–10.07) | 2.89 (0.95–8.91) | 3.15 (1.19–9.09) | 3.66 (0.90–9.36) | 4.33 (1.49–11.14) | 5.21 (1.43–12.53) | 6.15 (1.71–15.94) | 7.01 (3.12–16.19) |
| % of live births where mother <20 years old | 7.01 (1.14–14.50) | 6.82 (1.46–14.13) | 6.58 (1.35–12.82) | 6.44 (1.50–13.89) | 5.72 (1.17–11.89) | 5.39 (1.13–10.77) | 5.16 (1.30–9.93) | 4.33 (0.87–8.49) |
| % of live births where maternal LSOA history within the 10% most deprived LSOAs in England | 14.46 (0.00–60.18) | 14.98 (0.00–60.46) | 14.64 (0.00–61.92) | 15.33 (0.00–60.90) | 14.87 (0.00–60.36) | 14.75 (0.11–61.21) | 14.47 (0.00–61.52) | 14.72 (0.24–60.68) |
| % of live births where child has a congenital anomaly | 1.64 (0.60–3.34) | 1.64 (0.84–3.25) | 1.62 (0.60–2.74) | 1.78 (0.84–3.68) | 1.78 (0.93–3.19) | 1.81 (1.06–3.38) | 1.83 (1.11–3.61) | 1.93 (1.03–3.22) |
| % of live births with low birth weight | | | | | 6.26 (4.22–9.94) | | | |
| % of dependent child households with lone parent | | | | | 18.31 (9.78–30.94) | | | |
| Rate of violent crime (per 100 LA residents) | | | | | 1.14 (0.52–3.17) | | | |

The median LA value is presented as many of the explanatory measures are non-normally distributed.
ARA, adversity-related hospital admission; HES APC, Hospital Episode Statistics Admitted Patient Care; LA, local authority; LSOA, lower-layer super output area.

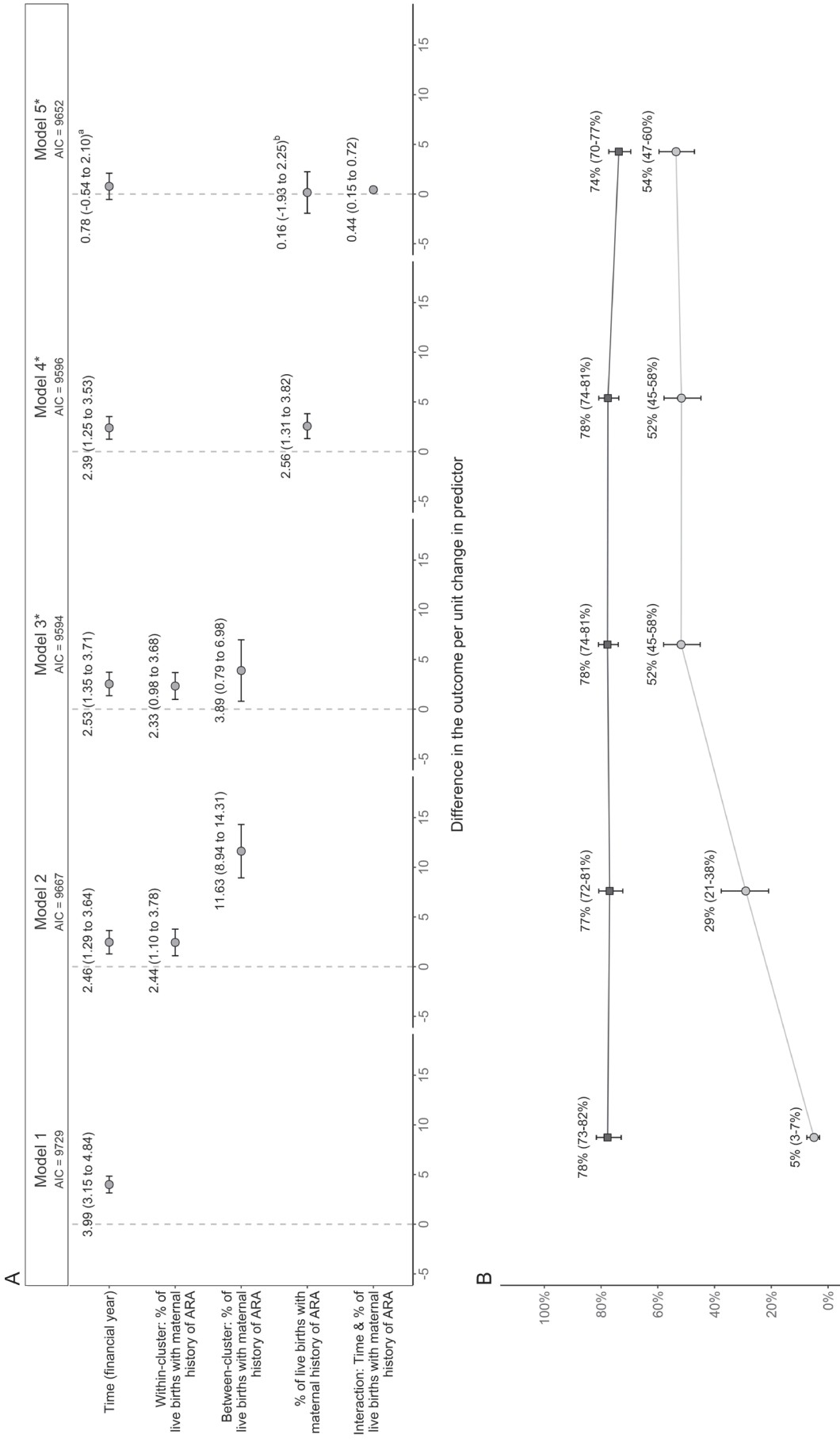

**Figure 2** (A) Modelling the association between LA-specific percentage of live births with maternal history of ARA and LA-specific rate of infant entry into care for 131 English LAs, over time (2006/2007 to 2013/2014). (B) Variation in the outcome explained by components of the models. *Models were adjusted for all other explanatory measures (table 1). The term 'fixed-effects' includes any explanatory measure in the model, such as time and maternal history of ARA, but does not include random effects such as random intercepts and random slopes; 95% CI given in brackets. [a]Effect where the percentage of live births with maternal history of ARA is equal to zero. [b]Effect in 2006/2007. AIC, Akaike information criteria; ARA, adversity-related hospital admission; LA, local authority.

covariates. Prior to adjustment (ie, model 2), a percentage point increase in percentage of live births with maternal history of ARA within the same LA was associated with an extra 2.44 infants, per 10 000, entering care (95% CI 1.10 to 3.78), while a percentage point increase between two different LAs was associated with an extra 11.63 infants, per 10 000, entering care (8.94 to 14.31). Using models 1 and 2, we estimated that the percentage of live births with a maternal history of ARA explained 24% (95% CI 14% to 35%) of the variation in the rate of infant entry into care between 2006/2007 and 2013/2014 (figure 2B). After adjustment for all other explanatory measures (ie, model 3), there was insufficient evidence that the effect on the outcome of increases to the percentage of live births with maternal history of ARA within the same LA and the effect on the outcome of increases between different LAs were different (p=0.36). After refitting the adjusted model without disaggregation of the within-LA and between-LA effects (ie, model 4), there was evidence that a 1% point increase in the percentage of live births with maternal history of ARA, either within the same LA or between two different LAs, was associated with an extra 2.56 infants per 10 000 entering care (1.31 to 3.82) over a 12-month period, holding all other model covariates constant. Finally, we explored whether the effect of an increase within the same LA or between two different LAs in percentage of live births with maternal history of ARA varied over the study period (ie, model 5). There was evidence that the magnitude of the association between the percentage of live births with maternal history of ARA (by LA, over the study period) and rate of infant entry into care increased over time between 2006/2007 and 2013/2014 (interaction coefficient estimate: 0.44, 95% CI 0.15 to 0.72), as seen in figure 3. (For full model results, see online supplementary appendix pages 8–11.)

## DISCUSSION

We found that increases either between LAs or within the same LA over time in the rate of infant entry into care were associated with an increased prevalence of maternal history of ARA. Evidence for this association persisted even after controlling for other, potentially confounding, LA-level risk factors for entry to care. The magnitude of the increase in rate of infant entry into care per percentage point increase in the percentage of live births with maternal history of ARA increased over time, particularly from 2009/2010 onwards (figure 3). We estimated that the percentage of live births with maternal history of ARA alone explained between 14% and 35% of the LA variation in rate of infant entry into care over the study period. The final model, with all covariates included, explained 47%–60% of this variation.

### Strengths

This is the first study to account for maternal health-related risk factors when examining variation among English LA rates of infant entries into care. A key strength of this study is the breadth of information on mothers and children included in the models. Six out of nine of our measures were derived using two national, longitudinal databases (HES APC and CLA), each with person-level records enabling follow-up through health and social care services throughout England over time. In particular, HES APC captures diagnoses via ICD-10 codes, allowing us to identify adversity-related healthcare need among mothers up to 3 years before delivery (ie, maternal history of ARA) that was sufficiently severe to be recorded during a hospital admission. Many of the ARA codes have been previously validated in other populations.[38–41] Another strength is the inclusion of multiple risk factors for maltreatment and infant entry into care in statistical modelling. We designed our modelling

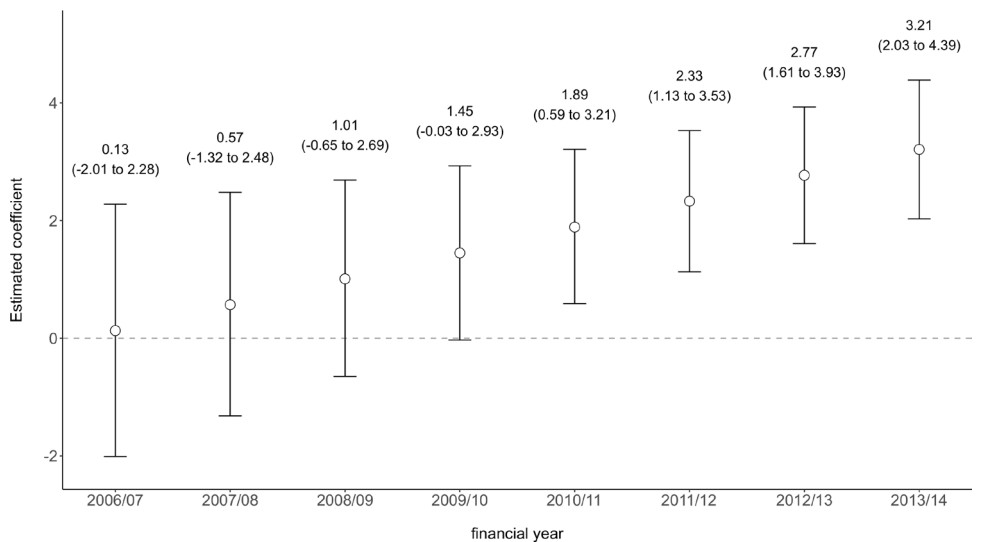

**Figure 3** Exploring changes in the association between the LA-specific percentage of live births with maternal history of ARA and LA-specific rate of infant entry into care between 2006/2007 and 2013/2014. ARA, adversity-related hospital admission; LA, local authority.

approach to balance model parsimony with adjustment for confounders that were supported by external evidence, which were relevant to policy and measurable. We further preserved model parsimony by allowing only our main exposure (maternal history of ARA) to vary over time, while fixing all other model covariates at their 2010/2011 values. Most of the explanatory measures in this final model were derived from administrative hospital records (HES APC), highlighting the importance of considering indicators for parental health near to birth when exploring LA variation in rate of infant entry to care.

## Limitations

The main limitation of this study sits with its ecological design. While we have found that increases in the percentage of live births with maternal history of ARA are associated with increases in the rate of infant entry to care over a given area, we cannot examine, at an individual level, whether children born to women with a history of adversity-related admission prior to birth were more likely to be placed into care during infancy. However, there is currently no English database containing mother-to-baby linked healthcare data with onward linkage to information on children's social care outcomes. We, therefore, studied the association between maternal adversity prior to birth and infant entry to care at an LA level. Another limitation is that we did not explore the effect of increases to LA prevalence of maternal history of hospital admissions related to particular types (or combination of types) of adversity prior to birth. We took this decision partly to avoid increasing the risk of type I error inflation due to excessive statistical testing (relative to our sample size) and partly due to a number of LAs having non-disclosable values (<10) for this measure when stratified by type (or combination of types) of adversity. There is also a lack of information on 'supply-side' factors to infant entry to care, such as funding for early intervention programmes and availability of foster care placements, of sufficient quality for research at the LA level for the whole of England. A further limitation is that we cannot distinguish whether increases over time in the percentage of live births with maternal history of ARA reflect a true increase or are partly explained by nationwide changes in coding practices, although adversity admissions appear to be increasing particularly among younger women.[42] Finally, we were limited to maternal and not paternal ARA as it is not possible to identify fathers in HES APC. We were also restricted to hospital indicators of adversity as primary care indicators cannot be linked to LA.

## Implications of the study findings

Until now, there has been a paucity of evidence on the association between local variation in demand for child protection intervention in England and parental health indicators that are associated with diminished parenting. Most prior studies have instead focused on quantifying the relationship between variation in demand

and poverty.[16 43 44] We found that hospital admissions related to adversity such as substance misuse, exposure to violence and mental health problems explained a substantial proportion of LA variation in the rate of infant entry into care, even after adjustment for maternal deprivation. The prevalence and contribution of this risk factor also appear to be increasing with time. These admissions present opportunities to respond to adversity-related healthcare need prior to pregnancy and birth to improve maternal health and potentially mitigate infant entry into care. However, the success of early intervention in a healthcare setting will rely on effective multiagency collaboration between health, social care and third-sector organisations within LAs, which was a key recommendation from the Care Crisis Review into the challenges facing the children's social care sector in England.[2] Services should also provide a relationship-based and flexible approach to support service users who initially fail to engage and to mitigate issues arising from distrust of professionals.[45 46] Further, parents and parents-to-be who are subject to children's social care involvement often have a relatively short window in which to make changes; therefore, services targeted to this population must offer timely access to ensure support is available within the timeframes of children's social care assessments and family court proceedings.

We saw increased acceleration over time in the average LA percentage of live births with maternal history of ARA and increased magnitude over time in the effect of increases to LA percentage of live births with history of maternal ARA on the rate of infant entry into care, which coincided with the introduction of cuts to central government funding for LAs.[47] These cuts led many LAs to decrease spending on early intervention, youth services and some public health programmes.[48 49] However, it was beyond the scope of this work to investigate the effect of austerity on changes to the association between the percentage of live births with maternal history of ARA and rate of infant entry into care over time.

This study highlights the importance of establishing a linked parent–child healthcare data resource with onward linkage to CLA and other social care data, to enable more robust evaluation of the association between maternal ARA and other health indicators and infant entry into care to inform preventive interventions. Such data linkages would be vital to inform policy strategies aimed at improving women's health, well-being and reproductive rights, and potentially reduce infant entries to care.

**Acknowledgements** We would like to thank Dr Ania Zylbersztejn and Dr Katie Harron for their past work on an extract of Hospital Episode Statistics Admitted Patient Care to identify congenital anomaly diagnoses among children and to link mother and child records, respectively. This work uses data provided by individuals and collected by local authorities, the Department for Education and the NHS as part of their care and support.

**Contributors** RJP carried out all data analysis and produced the first draft of this manuscript. RG contributed critically to the formulation of the research question and study design, with input from all other authors (RJP, MAJ, LPMMW, BDS, SS

and SJB). LPMMW and MAJ provided expertise on the HES APC and CLA data extracts, respectively. MAJ also reviewed content making reference to family law. SS reviewed the literature for validated ICD-10 codes to identify adversity-related healthcare use. BDS provided statistical guidance. All authors (RJP, MAJ, LPMMW, BDS, SS, SJB and RG) provided critical revision to the initial manuscript and approved the final version for publication.

**Funding** LPMMW, RG, RP and SJB are supported by funding from the Nuffield Foundation (grant reference number KID/42838). RG also receives funding from Health Data Research UK. MAJ is supported by funding from the Medical Research Council through the UCL Birkbeck Doctoral Training Partnership (MR/R502248/1). This work is supported by the NIHR GOSH BRC. This research benefits from and contributes to the NIHR Children and Families Policy Research Unit, but was not commissioned by the National Institute for Health Research (NIHR) Policy Research Programme. The views expressed are those of the author(s) and not necessarily those of the Nuffield Foundation, the NHS, the NIHR or the Department of Health.

**Competing interests** None declared.

**Patient consent for publication** Not required.

**Ethics approval** Both HES APC and CLA data were pseudonymised before we received them and all other data were deidentified or already publicly available and anonymised; therefore, this study did not require ethical approval.

**Provenance and peer review** Not commissioned; externally peer reviewed.

**Data availability statement** HES APC and the CLA return may be obtained from a third party (NHS Digital: https://digital.nhs.uk/services/data-access-request-service-dars; and the Department for Education: https://www.gov.uk/guidance/how-to-access-department-for-education-dfe-data-extracts, respectively) and are not publicly available. All other data used in this study are available in public, open access repositories: (1) https://www.ons.gov.uk/peoplepopulationandcommunity/populationandmigration/populationestimates, (2) https://www.nomisweb.co.uk/census/2011/lc1109ew, and(3) https://fingertips.phe.org.uk/search/violence#page/6/gid/1/pat/6/par/E12000008/ati/102/are/E06000039/cid/4.

**ORCID iDs**
Rachel Jane Pearson http://orcid.org/0000-0002-3644-2885
Matthew Alexander Jay http://orcid.org/0000-0003-2481-7755
Linda Petronella Martina Maria Wijlaars http://orcid.org/0000-0003-1222-2922

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
