## [Reviewer comments · BMJ Open]

ARTICLE DETAILS

TITLE (PROVISIONAL)	Association between health indicators of maternal adversity and the rate of infant entry to local authority care in England: a longitudinal ecological study
AUTHORS	Pearson, Rachel; Jay, Matthew; Wijlaars, Linda; De Stavola, Bianca; Syed, Shabeer; Bedston, Stuart; Gilbert, Ruth

VERSION 1 - REVIEW

REVIEWER	Julie Laurin Université de Montréal, Canada
REVIEW RETURNED	24-Feb-2020

GENERAL COMMENTS	I believe this paper has major potential for contributing to the literature. However, the analyses need major revisions, as well as their explanation in the discussion section. Please see my attached files for details. Within an ecological perspective, this paper assesses the association between prenatal maternal adversity and the rate of infant entry to local authority care in England. Using child, mother and neighbourhood level data within a longitudinal perspective, the authors tried to model the best predictors of infants' entry into local authority care. The addressed research question is formulated clearly and particularly innovative. This paper has the potential for great advancement in our understanding of pre- and post-natal risk factors of young families. Despite its many strengths, please find below the weakness I found in the paper. Introduction • The paper does not clearly explain the word "local authorities". Despite being a term commonly known in the UK, people in North America would have no reference point for it. Please add a definition to it.• Child abuse and neglect are linked to parental adversity, mostly as it depletes parents' self-regulatory abilities. Many predictors of better parental self-regulation are omitted from the analyses or not even mentioned in the introduction. For example, the existence of a coparent aiding at home, the number of weeks in paid parental leave following the birth taken by each parent, especially with respect to weeks where fathers are home alone with the child,
--

as well as access to subsidized childcare have each been shown to provide relief to mothers and aid in their adaptive functioning and parenting abilities.

- Many acronyms are used in the text rendering the text more confusing. Some are also never written out (e.g., 'ONS' p.6)
- Numbers including decimals are written weirdly, with the decimal point appearing in the middle instead of the bottom of the number. This presentation makes it harder to read and understand them.

Analysis

- My main apprehension for this article falls in the data analyses. It is a complex multilevel analysis, in which the authors, unfortunately, did not decompose each level of analysis.
 - o In this multilevel model, the authors included three levels of analysis, i.e., the child, the mother and the neighbourhood/environment. As the dependent variable is the probability of infants being admitted in local authority care, the first level of analysis is the child. As completed in this article, the first model needs to include only this variable in the initial model. However, a decomposition problem arises with the following models. The following models should respectively include the addition of a single independent variable at a time, beginning with child-level variables (i.e., % of live births with low birth weights and % of live births with complex chronic conditions), followed by mother-level variables (i.e., maternal age, maternal history of ARA), and then neighbourhood-level or environment-level variables (i.e., LA population size, % of dependent child households with lone parent, rate of violent crime, % of live births where maternal LSAO history within 10% most deprived). With each addition of a single independent variable in the model, the ΔR^2 need to be reported to provide an indication of additional variance explained from each.
 - o The authors should provide an explanation for their choice of LA-specific mean centering vs. the general mean-centering, as well as report their results according to the level of centering. For example, following an average prevalence of LA-specific maternal ARA compared to other LA-specific maternal ARA prevalence, the results indicate... vs. At an equal and average prevalence of maternal ARA, people living in a poorer neighborhood have a higher average level of ...).
 - o The wording "not disaggregated" is a double negative, which is poor writing. Please improve writing and clarify its meaning.
 - The reporting of the results and the presentation of the information in this paper needs to be improved. For example:
 - o At page 11, there is a large gap in the text, which needs to be rectified.
 - o The results should be reported in Z scores to improve comparisons of results. This transformation information should also be mentioned the text.
 - o The figures do not include titles. Also, a note should be added to ease understanding and readability of the reported results.
 - o In table 2, results are reported based on medians. Please provide means and standard deviation to improve readability and understanding of the results.
- #### Discussion
- An explanation for each decomposed model should be provided. Their interpretation should be easily understood from each descriptive explanation.
 - The results of the research are inadequately linked with previous literature.

	 The authors list as a strength the ability for exposure to maternal history ARA to vary over time while fixing all other covariates at their 2010-11 values. This choice was never properly explained in the methods and results section and is not further explained in the discussion. The authors need to provide an explanation for this choice, as well as explain it as a potential strength.
--	---

REVIEWER	Samantha Brown Colorado State University, USA
REVIEW RETURNED	22-Mar-2020

GENERAL COMMENTS	This study examined whether certain maternal adversity indicators in hospitalization and pre-delivery impact rate of infant entry into local authorities (LAs) in England. The manuscript is well-written and the study has many strengths, including the use of two longitudinal datasets and linear mixed-effects models. Overall the manuscript contributes to the literature. I have just a few suggestions to improve an already strong manuscript. Extant research on maternal history of adversity (i.e., adverse childhood experiences) shows that adversities often co-occur and that specific types may be implicated in differential outcomes. Since the authors had specific ICD-10 codes, why not delineate specific types or combinations of adversity and the impact on risk of entry into care? In addition, more discussion on this extant literature in relation to pathways to reentry into care in the introduction is warranted. In other countries, such as in the US, families are often disproportionately overrepresented in child protective services and health disparities exist among marginalized groups. Therefore, could race/ethnicity be a contributing factor in England and be controlled for in analyses? Can the authors clarify if infants entering care are the “target” child or could allegations of abuse and neglect been made toward an infant’s sibling, which resulted in an infant entering care? Since the authors derived a look-back period – three years prior to infant birth – it is unclear why the proportion of live births where the child was diagnosed with a complex chronic condition in early childhood was included as an explanatory factor. Because conditions could be diagnosed for a child up to 2-years old and thus surpasses the infant age into care (under 1 year), how might this impact results? Please clarify. In the modeling section the authors refer to time as financial year, but this isn’t specified in the earlier sections of the paper. It may be helpful to the reader to clarify the April-March yearly timeframe. Although the authors were not able to make direct linkages between a mother’s history of adversity and her own infant’s entry into care, findings inform policy and practice strategies that may be useful to mitigate risk of entry into care. Yet, the discussion is quite brief. The authors do not highlight what this study adds to the literature and how findings may inform more specific prevention and intervention
--

	programming and policy – this is a missed opportunity. Finally, it is likely that cuts to LA funding has significant impact on entry into care and should, at a minimum, be included in the limitations section of this paper.
--	---

VERSION 1 – AUTHOR RESPONSE

Reviewers' Comments to Author:

Reviewer: 1

Reviewer Name: Julie Laurin

Institution and Country: Université de Montréal, Canada Please state any competing interests or state

'None declared': none.

I believe this paper has major potential for contributing to the literature. However, the analyses need major revisions, as well as their explanation in the discussion section. Please see my attached files for details.

Within an ecological perspective, this paper assesses the association between prenatal maternal adversity and the rate of infant entry to local authority care in England. Using child, mother and neighbourhood level data within a longitudinal perspective, the authors tried to model the best predictors of infants' entry into local authority care. The addressed research question is formulated clearly and particularly innovative. This paper has the potential for great advancement in our understanding of pre- and post-natal risk factors of young families.

Thank you for this positive assessment of our paper and its potential impact.

Despite its many strengths, please find below the weakness I found in the paper.

Introduction

- The paper does not clearly explain the word "local authorities". Despite being a term commonly known in the UK, people in North America would have no reference point for it. Please add a definition to it.

Thank you for highlighting this, we have now added some context and a brief definition (page 4, line 16 of the tracked changes version of the revised manuscript):

'The responsibility for placing infants into care in England rests with the 152 local authorities (LAs) (i.e. local governmental bodies with responsibilities to provide community services, such as schools, housing, and social care) who have a statutory duty to safeguard children.'

- Child abuse and neglect are linked to parental adversity, mostly as it depletes parents' self-regulatory abilities. Many predictors of better parental self-regulation are omitted from the analyses or not even mentioned in the introduction. For example, the existence of a co-parent aiding at home, the number of weeks in paid parental leave following the birth taken by each parent, especially with respect to weeks where fathers are home alone with the child, as well as access to subsidized childcare have each been shown to provide relief to mothers and aid in their adaptive functioning and parenting abilities.

Thank you for highlighting these potential protective risk factors for infant entry into care.

We did not discuss all risk/protective factors for infant entry to care as this piece of work specifically focussing on the effect of a sole risk factor (maternal history of adversity-related hospital admissions prior to birth). We chose to focus on this exposure as it is potentially amenable to intervention through identification of maternal adversity in a healthcare setting before or during the early antenatal period. Therefore, our introduction centres on describing existing evidence on this pathway between maternal adversity prior to birth and infant maltreatment and entry into care rather than reviewing the evidence on all risk and protective factors for entry into care.

In addition, we do not have individual data for the predictors you have mentioned and there is also currently insufficient data on these factors at the local authority level for the whole of England.

- Many acronyms are used in the text rendering the text more confusing. Some are also never written out (e.g., 'ONS' p.6)

Thank you for pointing this out. We have removed some of the lesser used abbreviations (ONS and PHE) and are happy to take editorial direction as to the use of the remaining abbreviations in this paper. As our analysis pulls together many data sources, we had to strike a balance by reducing acronyms where possible but also by leaving those in place where acronyms are commonly used (such as ICD-10 codes, and data sets names i.e. CLA and HES APC) or where terms are long/unwieldy and referred to many times in the manuscript (such as maternal history of ARA and LSOA). We have also reviewed the manuscript to ensure all acronyms have been written out where first used (and that definitions of acronyms have been in the relevant table footnotes).

- Numbers including decimals are written weirdly, with the decimal point appearing in the middle instead of the bottom of the number. This presentation makes it harder to read and understand them.

We have now reformatted these.

Analysis

- My main apprehension for this article falls in the data analyses. It is a complex multilevel analysis, in which the authors, unfortunately, did not decompose each level of analysis.
 - In this multilevel model, the authors included three levels of analysis, i.e., the child, the mother and the neighbourhood/environment. As the dependent variable is the probability of infants being admitted in local authority care, the first level of analysis is the child. As completed in this article, the first model needs to include only this variable in the initial model. However, a decomposition problem arises with the following models. The following models should respectively include the addition of a single independent variable at a time, beginning with child-level variables (i.e., % of live births with low birth weights and % of live births with complex chronic conditions), followed by mother-level variables (i.e., maternal age, maternal history of ARA), and then neighbourhoodlevel or environment-level variables (i.e., LA population size, % of dependent child households with lone parent, rate of violent crime, % of live births where maternal LSAO history within 10% most deprived). With each addition of a single independent variable in the model, the ΔR^2 need to be reported to provide an indication of additional variance explained from each.

We have provided responses below to the each of the points made:

1) Regarding the levels of the analysis:

Although we do indeed include information on children, parents and the local area (local authority), these are all aggregated by local authority and financial year (Apr-Mar, 2006/07 to 2013/14) as this is

an ecological study. Therefore, our analysis has only two levels: the local authority and time. This is first described in Study design in the Methods section (Page 5, line 10).

2) Regarding the suggestion to include variables into the model one-by-one:

The aim of this analysis was to better understand the association between LA prevalence of maternal adversity (using a proxy measure derived from data on hospitalisations) and LA rate of infant entry into care. Therefore, our interest in other predictors of infant entry into care is limited to wanting to adjust for them to get unbiased estimates for our exposure of interest. It is therefore outside of the scope of this analysis to quantify the amount of variation in the outcome that each factor included in the model explains.

We have amended text in the Explanatory Measures section (page 5, line 32) to clarify this:

'We derived several other explanatory measures, selected to capture further 'demand-side' risk factors (i.e. local service need) for child maltreatment and entry into care, alongside our exposure of interest (Table 1). These measures were chosen based upon evidence,¹³⁻¹⁵ data availability and quality, and interpretability and were used to account for potential confounders in our statistical analyses.'

Further, there are 133 local authorities included in this analysis and 8 years of follow-up data. Therefore, there are 1064 (133 multiplied by 8) observations included in each model. This is by no means a small sample size, but neither is it large and it would increase our risk of finding spurious associations if we were to run a further eight models on our data in order to calculate difference in R squared values for each of the explanatory measures.

- The authors should provide an explanation for their choice of LA-specific mean centering vs. the general mean-centering, as well as report their results according to the level of centering. For example, following an average prevalence of LA-specific maternal ARA compared to other LA-specific maternal ARA prevalence, the results indicate... vs. At an equal and average prevalence of maternal ARA, people living in a poorer neighborhood have a higher average level of ...).

We have updated text regarding these methods to improve clarity and readability (page 10, line 6):

'LA-specific percentage of live births with maternal history of ARA was modelled as a time-varying covariate between 2006/07 and 2013/14 to allow us to examine whether the association under study changed over time. However, the estimated model coefficient for this time-varying covariate could be interpreted as either: (1) the effect of a unit increase in this covariate within the same LA or (2) the effect of a unit increase in this covariate across different LAs. Therefore, we sought to disaggregate these two effects by replacing the original LA-specific variable with two variables: (1) the mean percentage of live births with maternal history of ARA over the study period for each LA (i.e. the between-LA effect), and (2) the difference, within local authorities, between the original LA-specific yearly values and the LA-specific mean (i.e. the within-LA effect).'

- The wording "not disaggregated" is a double negative, which is poor writing. Please improve writing and clarify its meaning.

We have now updated this wording on page 13, line 40:

'After refitting the adjusted model without disaggregation of the within-LA and between-LA effects (i.e. Model 4), there was evidence that a 1% point increase in the percentage of live births with maternal history of ARA, either within the same LA or between two different LAs, was associated with an extra

2.56 infants per 10,000 entering care (1.31 to 3.82) over a 12-month period, holding all other model covariates constant.'

- The reporting of the results and the presentation of the information in this paper needs to be improved. For example:

- At page 11, there is a large gap in the text, which needs to be rectified.

This gap was caused by the necessitation to insert a large landscape table into the word document close to where it is first referenced, as per BMJ open author guidelines. This will be rectified should the paper be accepted and proceed to type setting.

- The results should, be reported in Z scores to improve comparisons of results. This transformation information should also be mentioned the text.

Thank you for your suggestion. We considered transformations when planning this analysis. However, by modelling the 'raw' rates, rather than standardised rates (i.e. by mean centring and dividing by the standard deviation), the model can be interpreted in the original units (e.g. number of infants entering care per 10,000 infants resident in the local authority and % of live births with maternal history of ARA), rather than as a change in terms in standard deviation. We feel that using the original units help us to ensure that the interpretation of our model results can be more easily understood by readers within health and social care policy and practice. In addition, the outcome already exhibits a normal distribution, which is a required assumption for our chosen linear modelling method. Because of these points, we feel there is no clear benefit to standardising these data.

- The figures do not include titles. Also, a note should be added to ease understanding and readability of the reported results.

The titles are provided in text and indicate where figures should be placed if accepted and published.

- In table 2, results are reported based on medians. Please provide means and standard deviation to improve readability and understanding of the results.

We chose to display median values in Table 2 as many of the variables shown in this table are non-normally distributed (i.e. they have a skewed distribution). Displaying the mean for the non-normally distributed variables will give a biased representation of the average LA values, as the mean will not represent the centre of the critical mass of values where data is skewed. In addition, the median and mean are approximately equal where data are normally distributed. We have added the following footnote into Table 2 (page 12):

'Note: the median LA value is presented as many of the explanatory measures are non-normally distributed.'

Discussion

- An explanation for each decomposed model should be provided. Their interpretation should be easily understood from each descriptive explanation.

The key results in this piece of work come from the final model. As we had limited space in the discussion, we highlight only the key findings. However, we interpret the results for each model in the results section and provide full model results in the appendix (please see page 13, line 28 for the relevant section).

- The results of the research are inadequately linked with previous literature.

We have adapted text in the Discussion -> Implications of the study findings section (page 15, line 23):

'Until now, there has been a paucity of evidence on the association between local variation in demand for child protection intervention in England and parental health indicators that are associated with

diminished parenting. Most prior studies have instead focussed on the quantifying the relationship between variation in demand and poverty.^{16,43,44} We found that hospital admissions related to adversity such as substance misuse, exposure to violence, and mental health problems explained a substantial proportion of LA variation in the rate of infant entry into care, even after adjustment for maternal deprivation. The prevalence and contribution of this risk factor also appears to be increasing with time. These admissions present opportunities to respond to adversity-related healthcare need prior to pregnancy and birth to improve maternal health and potentially mitigate infant entry into care. However, the success of early intervention in a healthcare setting will rely upon effective multi-agency collaboration between health, social care and third sector organisations within LAs, which was a key recommendation from the Care Crisis Review into the challenges facing the children's social care sector in England.² Services should also provide a relationship-based and flexible approach to support service users who initially fail to engage and to mitigate issues arising from distrust of professionals.^{45,46} Further, parents and parents-to-be who are subject to children's social care involvement often have a relatively short window in which to make changes; therefore, services targeted to this population must offer timely access to ensure support is available within the timeframes of children's social care assessments and family court proceedings.'

- The authors list as a strength the ability for exposure to maternal history ARA to vary over time while fixing all other covariates at their 2010-11 values. This choice was never properly explained in the methods and results section and is not further explained in the discussion. The authors need to provide an explanation for this choice, as well as explain it as a potential strength.

We chose to model our covariates in this manner to improve model parsimony and we mention this on page 10, line 20):

'All other LA-level risk factors for entry into care were included in models as non-time varying variables using data from 2010/11 only (i.e. midpoint of study period) as we observed minor variations and this improved model parsimony.'

We also mention this again later on as one of the key strengths of our analyses (page 14, line 33):
'We designed our modelling approach to balance model parsimony with adjustment for confounders that were supported by external evidence, which were relevant to policy and measurable. We further preserved model parsimony by allowing only our main exposure (maternal history of ARA) to vary over time, while fixing all other model covariates at their 2010/11 values.'

Reviewer: 2

Reviewer Name: Samantha Brown

Institution and Country: Colorado State University, USA Please state any competing interests or state 'None declared': None declared

This study examined whether certain maternal adversity indicators in hospitalization and pre-delivery impact rate of infant entry into local authorities (LAs) in England. The manuscript is well-written and the study has many strengths, including the use of two longitudinal datasets and linear mixed-effects models. Overall the manuscript contributes to the literature. I have just a few suggestions to improve an already strong manuscript.

Thank you for this very positive assessment of the study.

Extant research on maternal history of adversity (i.e., adverse childhood experiences) shows that adversities often co-occur and that specific types may be implicated in differential outcomes. Since

the authors had specific ICD-10 codes, why not delineate specific types or combinations of adversity and the impact on risk of entry into care?

Thank you for this comment. We did consider exploring differences in the association under study when restricting our exposure of interest to particular forms (or combinations of forms) of adversity when planning our analyses. However, we concluded that this further exploration was beyond the scope of this piece of work due to the generally low prevalence of maternal exposure to ARA. We have added the following sentence into our Limitations section to explain our decision (Page 15, line 8 of the tracked changes version of the revised manuscript):

'Another limitation is that we did not explore the effect of increases to LA prevalence of maternal history of hospital admissions related to particular types (or combination of types) of adversity prior to birth. We took this decision partly to avoid increasing the risk of Type I error inflation due to excessive statistical testing (relative to our sample size) and partly due to a number of LAs having non-disclosable values (< 10) for this measure when stratified by type (or combination of types) of adversity.'

In addition, more discussion on this extant literature in relation to pathways to reentry into care in the introduction is warranted.

In this piece of work, we focus on the relationship between maternal adversity and first entry into care in infancy rather than any entry (i.e. first and re-entries) and used data only on first entries into care for our outcome measure. However, we have added a sentence with references to literature summarising what is known about parental risk factors for re-entries to care/recurrent maltreatment allegations/repeated child protective service involvement (page 4, line 29:

'There is also evidence that children whose parents experience these forms of adversity are more likely to be subject to repeated child protection intervention.'

With reference to:

Jenkins BQ, Tilbury C, Hayes H, et al. Factors associated with child protection recurrence in Australia. *Child Abuse Negl* Published Online First: 2018. doi:10.1016/j.chiabu.2018.05.002

White OG, Hindley N, Jones DPH. Risk factors for child maltreatment recurrence: An updated systematic review. *Med. Sci. Law*. 2015. doi:10.1177/0025802414543855

In other countries, such as in the US, families are often disproportionately overrepresented in child protective services and health disparities exist among marginalized groups. Therefore, could race/ethnicity be a contributing factor in England and be controlled for in analyses?

This is a very pertinent point and one we considered in the planning stage of this study. Data on ethnicity in English hospitalisation records is generally poor (particularly within some local authorities as hospital coding practice varies considerably across England). Therefore, we had very poor information on maternal ethnicity. Further, we felt that the effect of ethnicity may be highly correlated with some of the other variables included in our analyses (i.e. maternal deprivation status). We also expect this relationship to vary within the available data on broad ethnic groupings in England, for example by religion and cultural practice – which we do not currently have appropriate measures for. In addition, previous research has shown that the scale of ethnic variation in rate of entries to care (among children of any age) in England is not as pronounced as in other countries (for example the US). (as found by Louise Mc Grath-Lone et al - <https://doi.org/10.1016/j.chiabu.2015.10.020>)

Can the authors clarify if infants entering care are the “target” child or could allegations of abuse and neglect been made toward an infant’s sibling, which resulted in an infant entering care?

In England, we unfortunately do not have information on relationships between children in the children’s social care data sets, including the CLA data set used in this analysis. We also have very limited information available on reason for entry into care (restricted to eight very vague categories where ‘abuse and neglect’ is most common). Due to these limitations, we were unable to identify siblings in the data or instances where infant entry was due to the circumstance you describe above. Therefore, our analysis includes all children who first entered care during infancy (i.e. regardless of whether they were the “target” child or not).

We have amended text in the Methods -> Outcome section to further clarify the data used in calculating the outcome measure (page 5, line 25):

‘We used a longitudinal extract of CLA, which contained episode-in-care-level information for all children who first entered care during infancy between 1st April 2005 and 31st March 2014, to derive our outcome. Infants who entered care for any reason were included, with the exception of infants who first entered care under respite arrangements (i.e. an agreed series of short-term breaks, typically provided for children with complex healthcare needs).’

In addition, a flow chart of the inclusion and exclusion criteria for this extract is included in the appendix (page 5, line 30):

‘For further information on this extract, see Appendix page 3.’

Since the authors derived a look-back period – three years prior to infant birth – it is unclear why the proportion of live births where the child was diagnosed with a complex chronic condition in early childhood was included as an explanatory factor. Because conditions could be diagnosed for a child up to 2-years old and thus surpasses the infant age into care (under 1 year), how might this impact results? Please clarify.

Thank you for bringing this to our attention. We incorrectly labelled this measure during the write-up – rather than estimating LA prevalence of babies who have any complex chronic conditions, this measure estimates LA prevalence of babies with congenital anomalies (which are a subgroup in the complex chronic condition code list used). The reasoning for follow-up time (rather than look-back) is to capture children born with congenital anomalies who did not have a congenital anomaly diagnosis recorded at birth or who were diagnosed later in early childhood. As congenital anomalies occur intrauterine, these conditions would be present at birth (whether diagnosed then or later).

We have corrected all mentions of this measure now. In particular, we have updated Table 1 on page 8 (and the accompanying footnote, on page 9):

Explanatory	% of live births where child has a complex chronic condition	2006/07 to 2013/14	% of singleton live births with mother-baby linkage where the child had a congenital anomaly - identified where a congenital anomaly- related ICD-10 code ^b was recorded in the child’s HES APC record within the first 2 years of life or recorded on a death certificate before the age of 5 years old (to capture children whose congenital anomaly diagnosis was not captured at birth or who were diagnosed later in life).	HES APC	Information on children with congenital anomalies was only available for births with mother-baby record linkage. Therefore this measure was calculated using only singleton live births with linkage available. A further nine LAs were excluded as they were missing mother-baby record linkage for more than 35% of singleton live births in at least one financial year between 2006/07 and 2013/14.
--	--------------------	---	---------	--

1

^b Diagnoses of congenital anomalies were identified using a subset of Feudtner et al’s ICD-10 code list (i.e. all Q codes).²⁹

In the modeling section the authors refer to time as financial year, but this isn't specified in the earlier sections of the paper. It may be helpful to the reader to clarify the April-March yearly timeframe. Thank you for highlighting this. We now have added in some explanation at the beginning of the Methods section (page 5, line 10):

'This longitudinal, ecological analysis used yearly aggregate measures between the 2006/07 and 2013/14 financial years (Apr-Mar) for 131 out of a possible 151 (as at April 2006) English LAs, derived from several data sources.'

Although the authors were not able to make direct linkages between a mother's history of adversity and her own infant's entry into care, findings inform policy and practice strategies that may be useful to mitigate risk of entry into care. Yet, the discussion is quite brief. The authors do not highlight what this study adds to the literature and how findings may inform more specific prevention and intervention programming and policy – this is a missed opportunity.

Thank you for raising this very important point. We have adapted text in the Discussion -> Implications of the study findings section (page 15, line 23):

'Until now, there has been a paucity of evidence on the association between local variation in demand for child protection intervention in England and parental health indicators that are associated with diminished parenting. Most prior studies have instead focussed on quantifying the relationship between geographic variation in rates of child protection intervention and poverty.^{16,43,44} We found that maternal hospital admissions before birth related to adversity such as substance misuse, exposure to violence, and mental health problems explained a substantial proportion of LA variation in the rate of infant entry into care, even after adjustment for maternal deprivation. The prevalence and contribution of this risk factor also appears to be increasing with time. These admissions present opportunities to respond to adversity-related healthcare need prior to pregnancy and birth to improve maternal health and potentially mitigate infant entry into care. However, the success of early intervention in a healthcare setting will rely upon effective multi-agency collaboration between health, social care and third sector organisations within LAs, which was a key recommendation from the Care Crisis Review into the challenges facing the children's social care sector in England.² Services should also provide a relationship-based and flexible approach to support service users who initially fail to engage and to mitigate issues arising from distrust of professionals.^{45,46} Further, parents and parents-to-be who are subject to children's social care involvement often have a relatively short window in which to make changes; therefore, services targeted to this population must offer timely access to ensure support is available within the timeframes of children's social care assessments and family court proceedings.'

Finally, it is likely that cuts to LA funding has significant impact on entry into care and should, at a minimum, be included in the limitations section of this paper.

We agree with the reviewer's comment. We have discussed this issue in the 'implications of the study findings' section within the discussion (Page 15, line 43):

'We saw increased acceleration over time in the average LA percentage of live births with maternal history of adversity-related hospital admission and increased magnitude over time in the effect of increases to LA percentage of live births with history of maternal adversity-related hospital admission on the rate of infant entry into care, which coincided with the introduction of cuts to central government funding for LAs.⁴¹ These cuts led many LAs to decrease spend on early intervention, youth services, and some public health programmes.^{42,43} However, it was beyond the scope of this work to investigate the effect of austerity on changes to the association between the percentage of live births with maternal history of ARA and rate of infant entry into care over time.'